# Theoretical Limits of Pipeline Parallel Optimization and Application to Distributed Deep Learning

**Igor Colin**  **Ludovic Dos Santos**  **Kevin Scaman**
Huawei Noah's Ark Lab

## Abstract

We investigate the theoretical limits of pipeline parallel learning of deep learning architectures, a distributed setup in which the computation is distributed *per layer* instead of *per example*. For smooth convex and non-convex objective functions, we provide matching lower and upper complexity bounds and show that a naive pipeline parallelization of Nesterov's accelerated gradient descent is optimal. For non-smooth convex functions, we provide a novel algorithm coined *Pipeline Parallel Random Smoothing* (PPRS) that is within a $d^{1/4}$ multiplicative factor of the optimal convergence rate, where $d$ is the underlying dimension. While the convergence rate still obeys a slow $\varepsilon^{-2}$ convergence rate, the depth-dependent part is accelerated, resulting in a near-linear speed-up and convergence time that only slightly depends on the depth of the deep learning architecture. Finally, we perform an empirical analysis of the non-smooth non-convex case and show that, for difficult and highly non-smooth problems, PPRS outperforms more traditional optimization algorithms such as gradient descent and Nesterov's accelerated gradient descent for problems where the sample size is limited, such as few-shot or adversarial learning.

## 1 Introduction

The recent advances in deep neural networks have brought these methods into indispensable work in previously hard-to-deal-with tasks, such as speech or image recognition. The ever growing number of samples available along with the increasing need for complex models have quickly raised the need of efficient ways of distributing the training of deep neural networks. Pipeline methods [1, 2, 3, 4, 5] are proven frameworks for parallelizing algorithms both from the samples and the parameters point of view. While several pipelining approaches for deep networks have arisen in the last few years, GPipe [1] offers a solid and efficient way of applying pipelining techniques to neural network training. In this framework, network layers are partitioned and training samples flow across them, only waiting for the next layer to be free, increasing the overall efficiency in a nearly linear way.

Although pipelining is essentially designed for tackling both parameters and samples distribution over a network, some specific fields such as few-shot learning, deep reinforcement learning or adversarial learning present imbalanced needs between data and model distribution. Indeed, these problems typically require the training of a large model with very few examples, thus encouraging the use of methods leveraging the information in each sample to its best potential. Randomized smoothing for machine learning [6, 7] evidenced a way of using data samples more efficiently. The overall idea is to replace the usual gradient information with an average of gradients sampled around the current parameter; this approach is particularly effective when dealing with non-smooth problems as it is equivalent to smoothing the objective function.

The objective of this paper is to provide a theoretical analysis of pipeline parallel optimization, and show that accelerated convergence rates are possible using randomized smoothing in this setting.

**Related work.**

Distributing the training of deep neural networks can be tackled from several angles. Data parallelism [2, 8] focused on distributing the mini-batches amongst several machines. This method is easy to implement but may show its limits when considering extremely complex models. Although some attempts have proven successful [9, 10], model parallelism is hard to adapt to the training of deep neural networks, due to the parameters interdependence. As a result, pipelining [1, 2, 3, 4, 5] offered a tailored approach for neural networks. Although these methods have been investigated for some time [3, 2], the recent advances in [1] evidenced a scalable pipeline parallelization framework.

Randomized smoothing applied to machine learning was first presented in [6]. In [7], this technique is used in a convex distributed setting, thus allowing the use of accelerated methods even for non-smooth problems and increasing the efficiency of each node in the network. While the landscape of neural networks with skip connections tends to be *nearly convex* around local minimums [11], and in several applications, including optimal control, neural networks may be engineered to be convex [12, 13, 14], the core of deep learning problems remains non-convex. Unfortunately, results about randomized sampling for non-convex problems [15, 16] are ill-suited for machine learning scenarios: linesearch is at the core of the method, requiring prohibitive evaluations of the objective functions at each step of the algorithm. The guarantees of [15] remain relevant for this work however, since they give a reasonable empirical criterion to consider when evaluating the different methods.

## 2 Pipeline parallel optimization setting

In this section, we present the pipeline parallel optimization problem and the types of operations allowed in this setting.

**Optimization problem.** We denote as *computation graph* a directed acyclic graph $G = (\mathcal{V}, \mathcal{E})$ containing containing only 1 leaf. Each root of $G$ represents input variables of the objective function, while the leaf represents the single (scalar) ouptut of the function. Let $n$ be the number of non-root nodes, $\Delta$ the *depth* of $G$ (i.e. the size of the largest directed path), and each non-root node $i \in [\![1, n]\!]$ is be associated to a function $f_i$ and a computing unit. We consider minimizing a function $f_G$ whose computation graph is $G$, in the following sense: $\exists (g_1, ..., g_n)$ functions of the input $x$ such that:

$$g_0(x) = x, \qquad g_n(x) = f_G(x), \qquad \forall i \in [\![1, n]\!], \ g_i(x) = f_i\Big((g_k(x))_{k \in \text{Parents}(i)}\Big), \quad (1)$$

where $\text{Parents}(i)$ are the parents of node $i$ in $G$ (see Figure 1).

We consider the following unconstrained minimization problem

$$\min_{\theta \in \mathbb{R}^d} \ f_G(\theta), \tag{2}$$

in a distributed setting. More specifically, we assume that each computing unit can compute a subgradient $\nabla f_i(\theta)$ of its own function in one unit of time, and communicate values (i.e. vectors in $\mathbb{R}^d$) to its neighbors in $G$. A *direct* communication along the edge $(i, j) \in \mathcal{E}$ requires a time $\tau \geq 0$. These actions may be performed asynchronously and in parallel.While pipeline-parallel optimization is an abstraction that may be used for many different distribution setups, one of the main application is DL architectures distributed on multiple GPUs (with memory limitations and communication bandwidths) by partitioning the model.

**Regularity assumptions.** Optimal convergence rates depend on the precise set of assumptions applied to the objective function. In our case, we will consider two different constraints on the regularity of the functions:

(A1) **Lipschitz continuity:** the objective function $f_G$ is $L$-Lipschitz continuous, in the sense that, for all $\theta, \theta' \in \mathbb{R}^d$,

$$|f_G(\theta) - f_G(\theta')| \leq L\|\theta - \theta'\|_2. \tag{3}$$

(A2) **Smoothness:** the objective function is differentiable and its gradient is $\beta$-Lipschitz continuous, in the sense that, for all $\theta, \theta' \in \mathbb{R}^d$,

$$\|\nabla f_G(\theta) - \nabla f_G(\theta')\|_2 \leq \beta\|\theta - \theta'\|_2. \tag{4}$$

Finally, we denote by $R = \|\theta_0 - \theta^*\|$ (resp. $D = f_G(\theta_0) - f_G(\theta^*)$) the distance (resp. difference in function value) between an optimum of the objective function $\theta^* \in \operatorname{argmin}_\theta f_G(\theta)$ and the initial value of the algorithm $\theta_0$, that we set to $\theta_0 = 0$ without loss of generality.

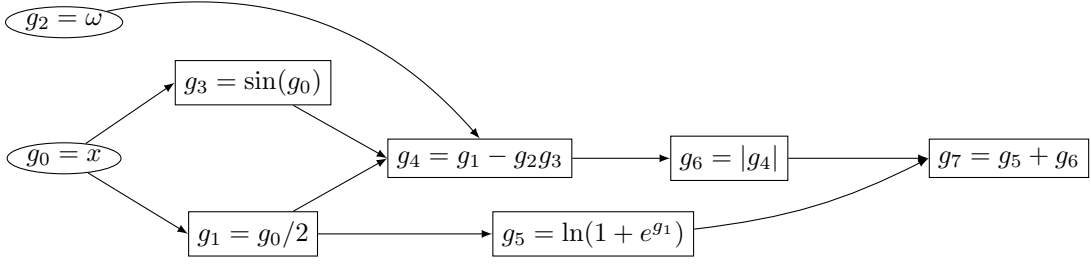

Figure 1: Example of a computation graph for $f(x, \omega) = \ln(1 + e^{x/2}) + |x/2 - \omega \sin(x)|$.

**Pipeline parallel optimization procedure.** Deep learning algorithms usually rely on backpropagation to compute gradients of the objective function. We thus consider first-order distributed methods that can access function values and matrix-vector multiplications with the Jacobian of individual functions (operations that we will refer to as *forward* and *backward* passes, respectively). A pipeline parallel optimization procedure is a distributed algorithm verifying the following constraints:

1. **Internal memory:** the algorithm can store past values in a (finite) internal memory. This memory may be shared in a central server or stored locally on each computing unit. For each computing unit $i \in [\![1, n]\!]$, we denote $\mathcal{M}_{i,t}$ the $d_i$-dimensional vectors in the memory at time $t$. These values can be accessed and used at time $t$ by the algorithm run by any computing unit, and are updated either by the computation of a local function (i.e. a forward pass) or a Jacobian-vector multiplication (i.e. a backward pass), that is, for all $i \in \{1, ..., n\}$,

$$\mathcal{M}_{i,t} \subset \mathrm{Span}\left(\mathcal{M}_{i,t-1} \cup \mathrm{FP}_{i,t} \cup \mathrm{BP}_{i,t}\right). \tag{5}$$

2. **Forward pass:** each computing unit $i$ can, at time $t$, compute the value of its local function $f_i(u)$ for an input vector $u = (u_1, ..., u_{|\mathrm{Parents}(i)|}) \in \prod_{k \in \mathrm{Parents}(i)} \mathcal{M}_{k,t-1}$ where all $u_i$'s are in the shared memory before the computation.

$$\mathrm{FP}_{i,t} = \left\{ f_i(u) \; : \; u \in \prod_{k \in \mathrm{Parents}(i)} \mathcal{M}_{k,t-1} \right\}. \tag{6}$$

3. **Backward pass:** each computing unit $j$ can, at time $t$, compute the product of the Jacobian of its local function $df_j(u)$ with a vector $v$, and split the output vector to obtain the partial derivatives $\partial_i f_j(u)v \in \mathbb{R}^{d_i}$ for $i \in \mathrm{Parents}(j)$.

$$\mathrm{BP}_{i,t} = \left\{ \partial_i f_j(u)v \; : \; j \in \mathrm{Children}(i), u \in \prod_{k \in \mathrm{Parents}(j)} \mathcal{M}_{k,t-1}, v \in \mathcal{M}_{j,t-1} \right\}. \tag{7}$$

4. **Output value:** the output of the algorithm at time $t$ is a $d_0$-dimensional vector of the memory,

$$\theta_t \in \mathcal{M}_{0,t}. \tag{8}$$

Several important aspects of the definition should be highlighted: 1) **Jacobian computation:** during the backward pass, all partial dervatives $\partial_i f_j(u)v \in \mathbb{R}^{d_i}$ for $i \in \mathrm{Parents}(j)$ are computed by the computing unit $j$ in a single computation. For example, if $f_j$ is a whole neural network, then these partial derivatives are all computed through a single backpropagation. 2) **Matrix-vector multiplications:** the backward pass only allows matrix-vector multiplications with the Jacobian matrices. This is a standard practice in deep learning, as Jacobian matrices are usually high dimensional, and matrix-matrix multiplication would incur a prohibitive cubic cost in the layer dimension (this is also the reason for the backpropagation being preferred to its alternative forward propagation to compute gradients of the function). 3) **Parallel computations:** forward and backward passes may be performed in parallel and asynchronously. 4) **Perfect load-balancing:** each computing unit is assumed to take the same amount of time to compute its forward or backward pass. This simplifying assumption is reasonable in practical scenarios when the partition of the neural network into local

functions is optimized through load-balancing [17]. 5) **No communication cost:** communication time is neglected between the shared memory and computing units, or between two different computing units. 6) **Simple memory initialization:** for simplicity and following [18, 7], we assume that the memory is initialized with $\mathcal{M}_{i,0} = \{0\}$.

## 3  Smooth optimization problems

Any optimization algorithm that requires one gradient computation per iteration can be trivially extended to pipeline parallel optimization by computing the gradient of the objective function $f_G$ sequentially at each iteration. We refer to these pipeline parallel algorithms as *naïve sequential extensions* (NSE). If $T_\varepsilon$ is the number of iterations to reach a precision $\varepsilon$ for a certain optimization algorithm, then its NSE reaches a precision $\varepsilon$ in time $O(T_\varepsilon \Delta)$, where $\Delta$ is the depth of the computation graph. When the objective function $f_G$ is smooth, we now show that, in a minimax sense, naïve sequential extensions of the optimal optimization algorithms are already optimal, and their convergence rate cannot be improved by more refined pipeline parallelization schemes.

### 3.1  Lower bounds

In both smooth convex and smooth non-convex settings, optimal convergence rates of pipeline parallel optimization consist in the multiplication of the depth of the computation graph $\Delta$ with the optimal convergence rate for standard single machine optimization.

**Theorem 1** (Smooth lower bounds)**.** *Let $G = (\mathcal{V}, \mathcal{E})$ be a directed acyclic graph of $n$ nodes and depth $\Delta$. There exists functions $f_i$ for $i \in [\![1, n]\!]$ such that $f_G$ is convex and $\beta$-smooth and reaching a precision $\varepsilon > 0$ with any pipeline parallel optimization procedure requires at least*

$$\Omega\left(\sqrt{\frac{\beta R^2}{\varepsilon}}\Delta\right).\tag{9}$$

*Similarly, there exists functions $f_i'$ for $i \in [\![1, n]\!]$ such that $f_G'$ is non-convex and $\beta$-smooth and reaching a precision $\varepsilon > 0$ with any pipeline parallel optimization procedure requires at least*

$$\Omega\left(\frac{\beta D}{\varepsilon^2}\Delta\right).\tag{10}$$

The proof of Theorem 1 relies on splitting the worst case function for smooth convex and non-convex optimization [19, 18, 20] so that it may be written as the composition of two well-chosen functions. Then, we show that any progress on the optimization requires to perform forward and backward passes throughout the entire computation graph, thus leading to a $\Delta$ multiplicative factor. The full derivation is available in the supplementary material.

The multiplicative factor $\Delta$ in these two lower bounds imply that, for smooth objective functions and under perfect load-balancing, *there is nothing to gain from pipeline parallelization*, in the sense that it is impossible to obtain sublinear convergence rates with respect to the depth of the computation graph, even when the computation of each layer is performed in parallel.

**Remark 1.** Note that our setting is rather generic and does not make any assumption on the form of the objective function. In more restricted settings (e.g. empirical risk minimization and objective functions that are averages of multiple functions, see Section 5), pipeline parallel algorithms may yet achieve substantial speedups (see for example GPipe for the training of deep learning architectures on large datasets [1]).

### 3.2  Optimal algorithm

Considering the form of the first and second lower bound in Theorem 1, naïve sequential extensions of, respectively, Nesterov's accelerated gradient descent for the convex setting and gradient descent for the non-convex setting lead to optimal algorithms [19]. Of course, this optimality is to be taken in a *minimax* sense, and does not imply that realistic functions encountered in machine learning cannot benefit from pipeline parallelization. However, this shows that one cannot prove better convergence rates for the class of smooth convex and smooth non-convex objective functions without adding additional assumptions. In the following section, we will see that non-smooth optimization leads to a more interesting behavior of the convergence rate and non-trivial optimal algorithms.

---

**Algorithm 1** Pipeline Parallel Random Smoothing

---

**Input:** iterations $T$, samples $K$, gradient step $\eta$, acceleration $\mu_t$, smoothing parameter $\gamma$.
**Output:** optimizer $y_T$
 1: $x_0 = 0$, $y_0 = 0$, $t = 0$
 2: **for** $t = 0$ to $T - 1$ **do**
 3:    Use pipelining to compute $g_k = \nabla f(x_t + \gamma X_k)$, where $X_k \sim \mathcal{N}(0, I)$ for $k \in [\![1, K]\!]$
 4:    $G_t = \frac{1}{K} \sum_k g_k$
 5:    $y_{t+1} = x_t - \eta G_t$
 6:    $x_{t+1} = (1 + \mu_t)y_{t+1} - \mu_t y_t$
 7: **end for**
 8: **return** $y_T$

---

## 4  Non-smooth optimization problems

For non-smooth objective functions, acceleration is possible, and we now show that the dependency on the depth of the computation graph only impacts a second order term. In other words, pipeline parallel algorithms can reduce the computation time and lead to near-linear speedups.

### 4.1  Lower bound

**Theorem 2** (Convex non-smooth lower bound). *Let $G = (\mathcal{V}, \mathcal{E})$ be a directed acyclic graph of $n$ nodes and depth $\Delta$. There exists functions $f_i$ for $i \in [\![1, n]\!]$ such that $f_G$ is convex and $L$-Lipschitz, and any pipeline parallel optimization procedure requires at least*

$$\Omega\left(\left(\frac{RL}{\varepsilon}\right)^2 + \frac{RL}{\varepsilon}\Delta\right) \tag{11}$$

*to reach a precision $\varepsilon > 0$.*

The proof of Theorem 2 relies on combining two worst-case functions of the non-smooth optimization literature: the first leads to the term in $\varepsilon^{-2}$, while the second gives the term in $\Delta\varepsilon^{-1}$. Similarly to Theorem 1, we then split these functions into a composition of two functions, and show that forward and backward passes are necessary to optimize the second function, leading to the multiplicative term in $\Delta$ for the second order term. The complete derivation is available in the supplementary material.

Note that this lower bound is tightly connected to that of non-smooth distributed optimization, in which the communication time only affects a second order term [7]. Intuitively, this effect is due to the fact that difficult non-smooth functions that lead to slow convergence rates are not easily separable as sums or compositions of functions, and pipeline parallelization can help in *smoothing* the optimization problem and thus improve the convergence rate.

### 4.2  Optimal algorithm

Contrary to the smooth setting, the naïve sequential extension of gradient descent leads to the suboptimal convergence rate of $O\left(\left(\frac{RL}{\varepsilon}\right)^2\Delta\right)$, which scales linearly with the depth of the computation graph. Following the *distributed random smoothing* algorithm [7], we apply random smoothing [6] to take advantage of parallelization and speedup the convergence. Random smoothing relies on the following smoothing scheme, for any $\gamma > 0$ and real function $f$, $f^\gamma(\theta) = \mathbb{E}\left[f(\theta + \gamma X)\right]$, where $X \sim \mathcal{N}(0, I)$ is a standard Gaussian random variable. This function $f^\gamma$ is a smooth approximation of $f$, in the sense that $f^\gamma$ is $\frac{L}{\gamma}$-smooth and $\|f^\gamma - f\|_\infty \leq \gamma L\sqrt{d}$ (see Lemma $E.3$ of [6]). Using an accelerated algorithm on the smooth approximation $f_G^\gamma$ thus leads to a fast convergence rate. Alg. 1 summarizes our algorithm, denoted *Pipeline Parallel Random Smoothing* (PPRS), that combines randomized smoothing [6] with a pipeline parallel computation of the gradient of $f_G$ similar to GPipe.[1] A proper choice of parameters leads to a convergence rate within a $d^{1/4}$ multiplicative factor of optimal.

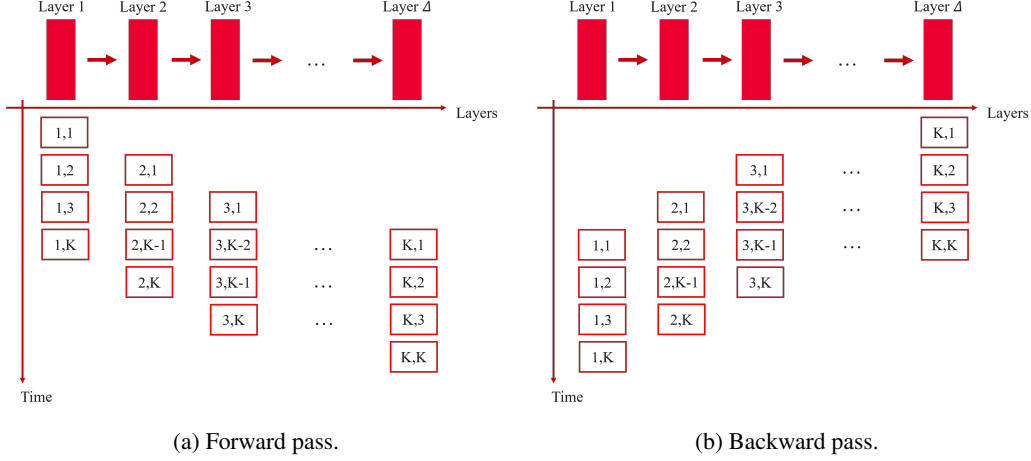

(a) Forward pass.

(b) Backward pass.

Figure 2: Bubbling scheme used at each iteration of PPRS in the case of a sequential neural network. Cell $(i, k)$ indicates the computation of the forward pass (resp. backward pass) for $\nabla f_i(\theta + \gamma X_k)$.

**Theorem 3.** *Let $f_G$ be convex and $L$-Lipschitz. Then, Alg. 1 with $K = \left\lceil (T+1)/\sqrt{d} \right\rceil$, $\eta = \frac{Rd^{-1/4}}{L(T+1)}$ and $\mu_t = \frac{\lambda_t - 1}{\lambda_{t+1}}$, where $\lambda_0 = 0$ and $\lambda_t = \frac{1+\sqrt{1+4\lambda_{t-1}^2}}{2}$, achieves an approximation error $\mathbb{E}\left[f_G(\theta_T)\right] - f_G(\theta^*)$ of at most $\varepsilon > 0$ in a time upper-bounded by*

$$O\left( \left( \frac{RL}{\varepsilon} \right)^2 + \frac{RL}{\varepsilon} \Delta d^{1/4} \right). \tag{12}$$

More specifically, Alg. 1 with $K$ gradient samples at each iteration and $T$ iterations achieves an approximation error of

$$\mathbb{E}\left[f_G(\theta_T)\right] - \min_{\theta \in \mathbb{R}^d} f_G(\theta) \leq \frac{3LRd^{1/4}}{T+1} + \frac{LRd^{-1/4}}{2K}, \tag{13}$$

where each iteration requires a time $2(K + \Delta - 1)$. When $K = T/\sqrt{d}$, we recover the convergence rate of Theorem 3 (the full derivation is available in the supplementary material).

**Randomized smoothing.** The PPRS algorithm described in Alg. 1 uses Nesterov's accelerated gradient descent [19] to minimize the smoothed function $f_G^\gamma$. This minimization is achieved in a *stochastic* setting, as the gradient $\nabla f_G^\gamma$ of the smoothed objective function is not directly observable. PPRS thus approximates this gradient by averaging multiple samples of the gradient around the current parameter $\nabla f_G(\theta + \gamma X_k)$, where $X_k$ are $K$ i.i.d Gaussian random variables.

**Gradient computation using pipeline parallelization.** As the random variables $X_k$ are independent, all the gradients $\nabla f_G(\theta + \gamma X_k)$ can be computed in parallel. PPRS relies on a *bubbling* scheme similar to the GPipe algorithm [1] to compute these gradients (step 3 in Alg. 1). More specifically, $K$ gradients are computed in parallel by sending the noisy inputs $\theta + \gamma X_k$ sequentially into the pipeline, so that each computing unit finishing the computation for one noisy input will start the next (see Figure 2). This is first achieved for the forward pass, then a second time for the backward pass, and thus leads to a computation time of $2(K + \Delta - 1)$ for all the $K$ noisy gradients at one iteration of the algorithm. Note that a good load balancing is critical to ensure that all computing units have similar computing times, and thus no straggler effect can slow down the computation.

**Non-convex case.** When the objective function is non-convex, randomized smoothing can still be used to smooth the objective function and obtain faster convergence rates. Unfortunately, the analysis of non-convex non-smooth first-order optimization is not yet fully understood, and the corresponding optimal convergence rate is, to our knownledge, yet unknown. We thus evaluate the non-convex version of PPRS in two ways:

1. We provide convergence rates for the averaged gradient norm used in [15], and prove that randomized smoothing can, as in the convex case, accelerate the convergence rate of non-smooth objectives to reach a smooth convergence rate in $\varepsilon^{-2}$.

2. We evaluate PPRS experimentally on the difficult non-smooth non-convex task of finding adversarial examples on CIFAR10 [21] with respect to the infinite norm (see Section 6).

While smooth non-convex convergence rates focus on the gradient norm, this quantity is ill-suited to non-smooth non-convex objective functions. For example, the function $f(x) = |x|$ leads to a gradient norm always equal to 1 (except at the optimum), and thus the convergence to the optimum does not imply a convergence in gradient norm. To solve this issue, we rely on a notion of average gradient used in [15] to analyse the convergence of non-smooth non-convex algorithms. We denote as $\bar{\partial}_r f(x)$ the Clarke $r$-subdifferential, i.e. the convex hull of all gradients of vectors in a ball of radius $r$ around $x$, that is $\bar{\partial}_r f(x) = \text{conv}\left(\{\nabla f(y) \ : \ \|y - x\|_2 \leq r\}\right)$, where $\text{conv}(A)$ is the convex hull of $A$. Then, we say that an algorithm reaches a gradient norm $\varepsilon > 0$ if the Clarke $r$-subdifferential contains a vector of norm inferior to $\varepsilon$, and $T_{r,\varepsilon} = \min\left\{t \geq 0 \ : \ \bar{\partial}_r f_G(\theta_t) \cap B_\varepsilon \neq \emptyset\right\}$, where $B_\varepsilon$ is the ball of radius $\varepsilon$ centered on 0. Informally, $T_{r,\varepsilon}$ is the time necessary for an algorithm to reach a point $\theta_t$ at distance $r$ from an $\epsilon$-approximate optimum. With this definition of convergence, PPRS converges with an accelerated rate of $\varepsilon^{-2}(\Delta + \varepsilon^{-2})$ (see supplementary material for the proof).

**Theorem 4.** *Let $f_G$ be non-convex and non-smooth. Then, Alg. 1 with $\gamma = \frac{r}{\sqrt{4\log(3L/\varepsilon)+2\log(2e)d}}$, $\eta = \gamma/L$, $\mu = 0$, $K = 18L^2/\varepsilon^2$ and $T = 36L(D + 2\gamma L\sqrt{d})/(\gamma\varepsilon^2)$ reaches a gradient norm $\varepsilon > 0$ in a time upper-bounded by*

$$T_{r,\varepsilon} \leq O\left(\frac{DL}{r\varepsilon^2}\left(\frac{L^2}{\varepsilon^2} + \Delta\right)\sqrt{d + \log\left(\frac{L}{\varepsilon}\right)}\right). \tag{14}$$

While the convergence rate in $\varepsilon^{-2}$ is indeed indicative of smooth objective problems, lower bounds are still lacking for this setting, and we thus do not know if $\varepsilon^{-4}$ is optimal for non-smooth non-convex problems. However, our experiments show that the method is efficient in practice on difficult non-smooth non-convex problems.

## 5 Finite sums and empirical risk minimization

A classical setting in machine learning is to optimize the empirical expectation of a loss function on a dataset. This setting, known as *empirical risk minimization* (ERM), leads to an optimization problem whose objective function is a finite sum

$$\min_{\theta \in \mathbb{R}^d} \ \frac{1}{m}\sum_{i=1}^{m} f_G(\theta, x_i), \tag{15}$$

where $\{x_i\}_{i \in [\![1,m]\!]}$ is the dataset. The main advantage of this formulation is that, to compute the gradient of the objective function, one must parallelize the computation of all the *per sample* gradients $\nabla_\theta f_G(\theta, x_i)$. GPipe takes advantage of this to parallelize the computation with respect to the examples [1]. While the naïve sequential extension of gradient descent achieves a convergence rate of $O\left(\left(\frac{RL}{\varepsilon}\right)^2 m\Delta\right)$, GPipe can reduce this by turning the product of $m$ and $\Delta$ into a sum: $O\left(\left(\frac{RL}{\varepsilon}\right)^2 (m + \Delta)\right)$. Applying the PPRS algorithm of Alg. 1 and parallelizing the gradient computations both with respect to the number of samples $K$ and the number of examples $m$ leads to a convergence rate of

$$O\left(\left(\frac{RL}{\varepsilon}\right)^2 m + \frac{RL}{\varepsilon}\Delta d^{1/4}\right), \tag{16}$$

which accelerates the term depending on the depth of the computation graph. This result implies that PPRS can outperform GPipe for ERM when the second term dominates the convergence rate, i.e. the number of training examples is smaller than the depth $m \ll \Delta$ and $d \ll (RL/\varepsilon)^4$. While these conditions are seldom seen in practice, they may however happen for few-shot learning and the fine tuning of pre-trained neural networks using a small dataset of task-specific examples.

## 6 Experiments

In this section, we evaluate PPRS against standard optimization algorithms for the task of creating adversarial examples. As discussed in Section 4, pipeline parallelization can only improve the

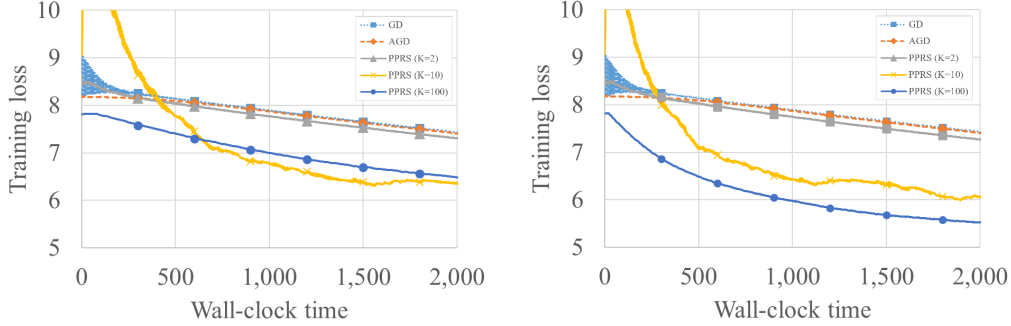

Figure 3: Comparison with GD and AGD. Increasing the number of samples increases the stability of PPRS and allows for faster convergence rates. Depth: (left) moderate, $\Delta = 20$. (right) high, $\Delta = 200$.

convergence of non-smooth problems. Our objective is thus to show that, for particularly difficult and non-smooth problems, PPRS can improve on standard optimization algorithms used in practice. We now describe the experimental setup used in our experiments on adversarial attack.

**Optimization problem:** We first take one image from one class of CIFAR10 dataset, change its class to another one and consider the minimization of the multi-margin loss with respect to the new class. We then add an $l_\infty$-norm regularization term to the noise added to the image. In other words, we consider the following optimization problem for our adversarial attack:

$$\min_{\tilde{x}} \sum_{i \neq y} \max\left\{0, 1 - f(\tilde{x})_y + f(\tilde{x})_i\right\} + \lambda\|\tilde{x} - x\|_\infty, \tag{17}$$

where $\|x\|_\infty = \max_i |x_i|$ is the $l_\infty$-norm, $x$ is the image to attack, $\tilde{x}$ is the attacked image, $y$ is the target class and $f$ is a pre-trained AlexNet [22]. The choice of the $l_\infty$-norm instead of the more classical $l_2$-norm is to create a difficult and highly non-smooth problem to better highlight the advantages of PPRS over more classical optimization algorithms.

**Parameters:** For all algorithms, we choose the best learning rates in lr $\in$ $\{10^{-3}, 10^{-4}, 10^{-5}, 10^{-6}, 10^{-7}\}$. For PPRS, we consider the following smoothing parameters $\gamma \in \{10^{-3}, 10^{-4}, 10^{-5}, 10^{-6}, 10^{-7}\}$ and investigate the effect of the number of samples by using $K \in \{2, 10, 100\}$. Following the analysis of Section 4.2, we do not accelerate the method and thus always choose $\mu = 0$ for our method. In practice, the accelerated version of the algorithm (with $\mu = 0.99$) did not improve the results. Hence, to improve the readability of the figures, we only focus on the (non-convex) theoretical version of the algorithm. We set $\lambda = 300$ and evaluate our algorithm in two parallelization settings: moderate ($\Delta = 20$) and high ($\Delta = 200$). Parallelization is simulated using a computation time of $2T(K + \Delta - 1)$ for an algorithm of $T$ iterations and $K$ gradients per iteration.

**Competitors:** We compare our algorithm with the standard gradient descent (GD) and Nesterov's accelerated gradient descent (AGD) with a range of learning rates and the standard choice of acceleration parameter $\mu = 0.99$.

Figure 3 shows the results of the minimization of the loss in Eq. (17) w.r.t. the number of epochs, averaged on 100 pairs of initial image and destination class. With a proper smoothing, PPRS significantly outperforms both GD and AGD. Moreover, increasing the number of samples increases the stability of PPRS and allows for faster convergence rates. For example, PPRS with a learning rate of $10^{-3}$ diverges for $K = 2$ but converges for $K = 10$ (the best learning rates are $10^{-4}$ for $K = 2$ and $10^{-3}$ for $K \in \{10, 100\}$). Moreover, GD and AGD require a smaller learning rate ($10^{-5}$ and $10^{-7}$, respectively) to converge, which leads to slow convergence rates. Note that, while the non-convexity of the objective function implies that multiple local minimums may exists and all algorithms may not converge to the same value, the *speed* of convergence of PPRS is higher than its competitors. Convergence to better local minimums due to a smoothing effect of the method are interesting research directions that are left for future work.

# 7 Conclusion

This work investigates the theoretical limits of pipeline parallel optimization by showing that, in such a setting, only non-smooth problems may benefit from parallelization. These hard problems can be accelerated by smoothing the objective function. We show both theoretically and in practice that such a smoothing leads to accelerated convergence rates, and may be used for settings where the sample size is limited, such as few-shot or adversarial learning. The design of practical implementations of PPRS, as well as adaptive methods for the choice of parameters $(K, \gamma, \eta, \mu)$ are left for future work.

## Footnotes

[1]GPipe (with GD/AGD) may be seen as a special case of PPRS, with $\gamma, K = 0$ (i.e., no randomized smoothing).

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
