[Supplementary Material · deep_distr_op_supplementary.pdf]

# Theoretical Limits of Pipeline Parallel Optimization and Application to Distributed Deep Learning SUPPLEMENTARY MATERIAL

## Abstract

This supplementary document contains complete proofs of the theorems presented in the article "Theoretical Limits of Pipeline Parallel Optimization and Application to Distributed Deep Learning".

## 1 Proofs of lower bounds

All proofs of lower bounds rely on splitting the worst-case functions of convex and non-convex optimization [1, 2, 3].

**Convex and smooth case** Let $\beta > 0$, $\mathcal{G}$ a computation graph and $i_1, ..., i_\Delta \subset [\![1, n]\!]$ a chain of non-root nodes of size $\Delta$. Let the functions $f_j : \mathbb{R}^d \to \mathbb{R}$ for $j \in [\![1, n]\!]$ be defined as

- $f_{i_1}(\theta) = \left(\theta, \frac{\beta}{8}\left[\sum_{i=1}^{k}(\theta_{2i+1} - \theta_{2i})^2 + \theta_1^2 - 2\theta_1\right]\right) \in \mathbb{R}^{d+1}$

- $f_{i_\Delta}(\theta) = \theta_{d+1} + \frac{\beta}{8}\left[\sum_{i=1}^{k}(\theta_{2i} - \theta_{2i-1})^2 + \theta_{2k+1}^2\right]$

- $f_{i_j}(\theta) = \theta$ if $j \in [\![1, \Delta - 1]\!]$

- $f_j(\theta) = 0$ otherwise

where $k \in \mathbb{N}$ is a parameter of the function and all functions $f_{i_k}$ in the chain $\{i_1, ..., i_\Delta\}$ only depend on their predecessor's output $\theta_{i_{k-1}}$. Intuitively, a partial sum is stored in the $d + 1$ coordinate and updated on node $i_1$ and $i_\Delta$. By construction, the objective function is

$$f_G(\theta) = \frac{\beta}{8}\left[\sum_{i=1}^{2k}(\theta_{i+1} - \theta_i)^2 + \theta_1^2 + \theta_{2k+1}^2 - 2\theta_1\right]. \tag{1}$$

First, note that $\nabla f_G(\theta) = \frac{\beta}{8}(M\theta - 2e_1)$ where $M = \begin{pmatrix} M' & 0 \\ 0 & 0 \end{pmatrix}$ and $M' \in \mathbb{R}^{(2k+1) \times (2k+1)}$ is a tridiagonal matrix with 2 on the diagonal and $-1$ on the upper and lower diagonals. A simple calculation shows that $0 \preceq M \preceq 4I$, and thus $f_G$ is $\beta$-smooth. The optimum of $f_G$ is obtained for $\theta_i^* = 1 - \frac{i}{2k+2}$, and

$$f_G(\theta^*) = -\frac{\beta}{8}\left(1 - \frac{1}{2k+2}\right). \tag{2}$$

Let $k_t = \max_i |\{k \in [\![1, d]\!] : \exists \theta \in \mathcal{M}_{i,t} \text{ s.t. } \theta_k \neq 0\}|$ be the maximum number of non-zero coordinates between 1 and $d$. Due to the form of the local functions, forward passes cannot increase $k_t$. Moreover, each backward pass can only increase the number of non-zero coordinates by one: on

node $i_1$ for odd number of coordinates, and on node $i_\Delta$ for even number of coordinates. Hence, one can only increase the number of non-zero coordinates by performing a backward pass on $i_1$, then a forward pass on $i_1, ..., i_{\Delta-1}$, and finally a backward pass on $i_{\Delta-1}, ..., i_2$ in order to increase the number of non-zero coordinates and send it to node $i_1$. When $\Delta \geq 2$, this leads to at least $\Delta - 1$ operations to increase $k_t$ by one, and thus

$$k_t \leq \left\lfloor \frac{t-1}{\Delta-1} \right\rfloor + 1 \leq \frac{2(t-1)}{\Delta} + 1. \tag{3}$$

Moreover, the last upper bound of Eq. (3) also holds when $\Delta = 1$, as we then have $k_t \leq \lfloor t \rfloor$. Finally, for all $i \in [\![k_t, d]\!]$, one has $\theta_{t,i} = 0$ and

$$f_G(\theta_t) \geq -\frac{\beta}{8} \left( 1 - \frac{1}{k_t + 1} \right). \tag{4}$$

Choosing $k = k_t$ and noting that $R^2 = \|\theta_0 - \theta^*\|^2 = \|\theta^*\|^2 \leq \frac{2(k+1)}{3}$ directly implies

$$f_G(\theta_t) - f_G(\theta^*) \geq \frac{\beta}{16(k_t+1)} \geq \frac{3\beta R^2}{32 \left( \frac{2(t-1)}{\Delta} + 2 \right)^2}. \tag{5}$$

**Non-convex and smooth case**   Let $\beta > 0$, $\mathcal{G}$ a computation graph and $i_1, ..., i_\Delta \subset [\![1, n]\!]$ a chain of non-root nodes of size $\Delta$. Let the functions $f_j : \mathbb{R}^d \to \mathbb{R}$ for $j \in [\![1, n]\!]$ be defined as

- $f_{i_1}(\theta) = \left( \theta, -\Psi(1)\Phi(\theta_1) + \sum_{i=1}^k \Psi(-\theta_{2i})\Phi(-\theta_{2i+1}) - \Psi(\theta_{2i})\Phi(\theta_{2i+1}) \right) \in \mathbb{R}^{d+1}$

- $f_{i_\Delta}(\theta) = \theta_{d+1} + \sum_{i=1}^k \Psi(-\theta_{2i-1})\Phi(-\theta_{2i}) - \Psi(\theta_{2i-1})\Phi(\theta_{2i})$

- $f_{i_j}(\theta) = \theta$ if $j \in [\![1, \Delta - 1]\!]$

- $f_j(\theta) = 0$ otherwise

where $k \in \mathbb{N}$ is a parameter of the function, $\Psi(x) = \mathbb{1}\{x > 1/2\} \exp\left( 1 - (2x-1)^{-2} \right)$, $\Phi(x) = \sqrt{e} \int_{-\infty}^x \exp(-t^2/2)dt$, and all functions $f_{i_k}$ in the chain $\{i_1, ..., i_\Delta\}$ only depend on their predecessor's output $\theta_{i_{k-1}}$. By construction, the objective function is

$$f_G(\theta) = -\Psi(1)\Phi(\theta_1) + \sum_{i=1}^{2k} \Psi(-\theta_i)\Phi(-\theta_{i+1}) - \Psi(\theta_i)\Phi(\theta_{i+1}). \tag{6}$$

This function was used in [3] to prove lower bounds on the convergence rate of non-convex smooth optimization. Moreover, similarly to the convex case, the number of non-zero coordinates can only increase when performing a backward pass on $i_1$ (for odd number of coordinates) and $i_\Delta$ (for even number of coordinates). Hence, using [3, Theorem 1] and Eq. (3), we have that, for any black-box optimization procedure, the time to reach a precision $\varepsilon > 0$ is lower bounded by

$$T_\varepsilon \geq 1 + \frac{\Delta}{2}(k_t - 1) \geq 1 + \frac{\Delta}{2} \left( \frac{\beta D}{c\varepsilon^2} - 1 \right), \tag{7}$$

where $c$ is a constant and $D = f_G(\theta_0) - \min_\theta f_G(\theta)$ is the initial distance to optimum in function value.

**Convex and non-smooth case**   Let $L > 0$, $\mathcal{G}$ a computation graph and $i_1, ..., i_\Delta \subset [\![1, n]\!]$ a chain of non-root nodes of size $\Delta$. Let the functions $f_j : \mathbb{R}^d \to \mathbb{R}$ for $j \in [\![1, n]\!]$ be defined as

- $f_{i_1}(\theta) = \left( \theta, \gamma \sum_{i=1}^k |\theta_{2i} - \theta_{2i-1}| + \delta \max_{i \in \{2k+2, ..., 2k+1+l\}} \theta_i \right) \in \mathbb{R}^{d+1}$

- $f_{i_\Delta}(\theta) = \theta_{d+1} + \gamma \sum_{i=1}^k |\theta_{2i+1} - \theta_{2i}| - \beta\theta_1 + \frac{\alpha}{2}\|\theta\|_2^2$

- $f_{i_j}(\theta) = \theta$ if $j \in [\![1, \Delta - 1]\!]$

- $f_j(\theta) = 0$ otherwise

where $\gamma, \delta, \beta, \alpha > 0$ and $k, l \geq 0$ are parameters of the function satisfying $2k + l < d$. The objective function is thus

$$f_G(\theta) = \gamma \sum_{i=1}^{2k} |\theta_{i+1} - \theta_i| - \beta\theta_1 + \delta \max_{i \in \{2k+2,...,2k+1+l\}} \theta_i + \frac{\alpha}{2} \|\theta\|_2^2. \quad (8)$$

This is the function used in [4, Theorem 2] to prove non-smooth convex lower bounds for distributed optimization, and the proof is identical by replacing $k_t$ the number of non-zero coordinates at time $t$ by its correct value given in Eq. (3). Thus, we have, for $t < \min\{l, k\Delta\}$,

$$f_G(\theta_t) - f_G(\theta^*) \geq \frac{1}{2\alpha n} \left[ \frac{\gamma^2}{2k} + \frac{\delta^2}{l} \right]. \quad (9)$$

Setting $\beta = \gamma(1 + \frac{1}{\sqrt{2k}})$, $\delta = \frac{L}{9}$, $\gamma = \frac{L}{9\sqrt{k}}$, $l = \lfloor t \rfloor + 1$, and $k = \lfloor \frac{t}{\Delta} \rfloor + 1$ leads to $t < \min\{l, k\Delta\}$ and

$$f_G(\theta_t) - f_G(\theta^*) \geq \frac{RL}{36} \sqrt{\frac{1}{(1 + \frac{t}{\Delta})^2} + \frac{1}{1+t}}, \quad (10)$$

while $f_G$ is $L$-Lipschitz and $\|\theta^*\|_2 \leq R$. Inverting this inequality leads to the desired bound on the time to reach a fixed precision.

## 2  Proofs of PPRS convergence rates

The PPRS algorithm uses Nesterov's accelerated gradient descent on the $L/\gamma$-smooth function $f_G^\gamma$. In order to use a single algorithm for both convex and non-convex settings, we did not use the off-the-shelf random smoothing algorithm of [5] that is specifically tailored to convex settings. Moreover, the simplicity, wide use and good performances of accelerated gradient descent in the deep learning community makes it a good candidate for real practical scenarios.

**Convex case**   To prove the convergence of PPRS for convex objective functions, we a convergence result for accelerated gradient descent in the presence of noise on the gradient.

**Lemma 1.** *Let $f : \mathbb{R}^d \to \mathbb{R}$ be a $\beta$-smooth convex function and $g_t$ a stochastic gradient of $f$ such that $\mathbb{E}[g_t] = \nabla f(x_t)$ and $\mathrm{var}(g_t) \leq \sigma^2$. Then, Nesterov's accelerated gradient descent with $\eta = 1/\beta$ and $\mu_t = \frac{\lambda_t - 1}{\lambda_{t+1}}$, where $\lambda_0 = 0$ and $\lambda_t = \frac{1 + \sqrt{1 + 4\lambda_{t-1}^2}}{2}$, leads to an approximation error*

$$f(y_t) - f(x^*) \leq \frac{2\beta\|x_0 - x^*\|^2}{(t+1)^2} + \frac{(t+1)\sigma^2}{2\beta}, \quad (11)$$

*where $x^*$ is a minimizer of the objective function $f$.*

*Proof.* This is a direct extension of the proof of [2, Theorem 3.19] to the case of stochastic gradients. $\qquad\square$

Applying Lemma 1 to the optimization of $f_G^\gamma$ leads to an approximation

$$f_G(\theta_T) - f_G(\theta^*) \leq \frac{2LR^2}{\gamma(T+1)^2} + \frac{(T+1)\sigma^2\gamma}{2L} + L\gamma\sqrt{d}, \quad (12)$$

where $\theta^*$ is a minmizer of $f_G$. Finally, since $\sigma \leq L/\sqrt{K}$, choosing $\gamma = \frac{Rd^{-1/4}}{T+1}$ and $K = \lceil (T+1)/\sqrt{d} \rceil$ leads to, after $T$ iterations,

$$f_G(\theta_T) - f_G(\theta^*) \leq \frac{3LRd^{1/4}}{T+1} + \frac{LRd^{-1/4}}{2K} \leq \frac{7}{2} \cdot \frac{LRd^{1/4}}{T+1}. \quad (13)$$

Since, each iteration takes a time $2(K - \Delta + 1)$, we thus reach a precision $\varepsilon$ in time

$$2T(K + \Delta - 1) \leq \frac{2T(T+1)}{\sqrt{d}} + 2T\Delta \leq \frac{49}{2} \left( \frac{LR}{\varepsilon} \right)^2 + \frac{7LR}{\varepsilon} \Delta d^{1/4}. \quad (14)$$

**Non-convex case** First, we use a convergence rate of gradient decent in the presence of additive noise.

**Lemma 2.** *Let $f : \mathbb{R}^d \to \mathbb{R}$ be convex and $\beta$-smooth, and $g_t$ be a noisy gradient of $f$, i.e. $\mathbb{E}\left[g_t\right] = \nabla f(\theta_t)$ and $\mathrm{var}(g_t) \leq \sigma^2$. Then, gradient descent with $\eta = 1/\beta$ leads to*

$$\min_{t \leq T} \|\nabla f(\theta_t)\|^2 \leq \frac{2\beta(f(\theta_0) - f(\theta^*))}{T} + \sigma^2 \,. \tag{15}$$

*Proof.* Using the smoothness, of $f$, we have

$$\begin{aligned} f(\theta_{t+1}) &\leq f(\theta_t) + \nabla f(\theta_t)^\top(\theta_{t+1} - \theta_t) + \tfrac{\beta}{2}\|\theta_{t+1} - \theta_t\|^2 \\ &\leq f(\theta_t) - \tfrac{1}{2\beta}\|\nabla f(\theta_t)\|^2 + \tfrac{1}{2\beta}\sigma^2 \,, \end{aligned} \tag{16}$$

and thus

$$\|\nabla f(\theta_t)\|^2 \leq 2\beta(f(\theta_t) - f(\theta_{t+1})) + \sigma^2 \,. \tag{17}$$

Summing over all times $t \leq T$ gives

$$\min_{t \leq T} \|\nabla f(\theta_t)\|^2 \leq \frac{1}{T}\sum_{t \leq T}\|\nabla f(\theta_t)\|^2 \leq \frac{2\beta(f(\theta_0) - f(\theta^*))}{T} + \sigma^2 \,. \tag{18}$$

$\square$

Applying Lemma 2 to the minimization of $f_G^\gamma$ gives

$$\min_{t \leq T} \|\nabla f_G^\gamma(\theta_t)\|^2 \leq \frac{2L(f_G(\theta_0) - f_G(\theta^*) + 2\gamma L\sqrt{d})}{\gamma T} + \frac{L^2}{K} \,. \tag{19}$$

Finally, we use a tail bound for the norm of Gaussian random variables and the fact that, by definition of $\bar{\partial}_r f_G(\theta_t)$, we have that

$$v = \mathbb{E}\left[\nabla f_G(\theta_t + \gamma X) \mid \|X\| \leq a\sqrt{d}\right] \in \bar{\partial}_{a\sqrt{d}\gamma} f_G(\theta_t) \,, \tag{20}$$

where $a \geq 1$. More specifically, we have

$$\|\nabla f_G^\gamma(\theta_t)\| \geq (1 - p_a)\|v\| - p_a L \,, \tag{21}$$

where $p_a = \mathbb{P}\left(\|X\| > a\sqrt{d}\right) \leq (a^2 e^{1-a^2})^{d/2} \leq (2e)^{d/2} e^{-da^2/4}$ using Chernoff's bound on a Chi-square random variable. The result then follows by inverting Eq. (21) and replacing $\|\nabla f_G^\gamma(\theta_t)\|$ by its upper bounds

$$\|v\| \leq \frac{1}{1 - p_a}\left(p_a L + \sqrt{\frac{2L(f_G(\theta_0) - f_G(\theta^*) + 2\gamma L\sqrt{d})}{\gamma T} + \frac{L^2}{K}}\right) \,, \tag{22}$$

and choosing $\gamma = \frac{r}{\sqrt{4\log(3L/\varepsilon) + 2\log(2e)d}}$, $K = \frac{18L^2}{\varepsilon^2}$ and $T = \frac{36L(D + 2\gamma L\sqrt{d})}{\gamma\varepsilon^2}$ gives $\|v\| \leq \varepsilon$, which implies that

$$T_{r,\varepsilon} \leq 2T(K + \Delta - 1) = O\left(\frac{DL}{r\varepsilon^2}\left(\frac{L^2}{\varepsilon^2} + \Delta\right)\sqrt{d + \log\left(\frac{L}{\varepsilon}\right)}\right) \,. \tag{23}$$