[Reviews · NeurIPS 2019]

Reviewer 1



Originality: The method (pipelining + smoothing) is derivative. At the same time, I am not certain if the proof techniques are original or not. Overall, I would consider the paper to have low-to-medium originality. Quality: The paper presents a thorough and important collection theoretical results, and provides excellent insight into the problem setting. There are some issues with the way the empirical results are presented (see the improvements section). Overall medium-high quality. Clarity: I found the paper well-written and easy to follow. High clarity. Significance: Although the authors were mostly concerned with the non-smooth, low-sample setting, I find that the paper addresses an important intersection of topics: ML-systems (pipelining), optimization theory, and deep learning. The intersection of these topics will only continue to grow in importance, and work such as this paper is highly significant.

Reviewer 2



**Update after reading the author's feedback With the additional information on connections with existing work and comments and the practical setup, I'm willing to update the score to 7. ******************************************************* The characterization of the distributed deep learning in the form of pipeline optimization seems to be a novel contribution, and the convergence results, particularly incorporated with randomized smoothing look reasonable. Some comments regarding clarifying the contribution: 1. The relationship between the proposed pipeline parallel optimization setting and existing work is not clear. Does it contain related work as special cases? The authors mentioned in the abstract that the presented study is distributed per-layer instead of per-sample. It could be helpful to give additional comparison along this line. 2. The manuscript seems to be short of details on the distributed computing mechanism. This was briefly touched in Section 2 on asynchronous value/gradient evaluation. Additional discussions such like the distributed framework, scalability etc. could add more practical value to the submission. This part is also unclear in the evaluation section of the paper. The improvement discussed in Section 5 over GPipe shows interesting trade-off, however as the authors mentioned those conditions are seldom seen in practice and the experiment setup seems artificial. 3. In the beginning of Section 4, the authors mentioned acceleration is possible. What’s the counterpart that the method is evaluated against? While the manuscript is an interesting read from the theoretical perspective, the reviewer is interested to see additional evidence on the practical impact such as improvement over state-of-the-art methods on well-studied applications.

Reviewer 3



I read the author's response which addresses the raised concerns, esp. regarding the general applications and the shown experimental results. I raise my rating to 7. The theoretical findings and contributions of this work are of general significance and give a nice overview of pipeline parallel optimization for different classes of functions. Further, the introduced PPRS optimization algorithm for non-smooth (and potentially almost non-smooth, e.g. L >>) functions is of general interest to the ML / DL community. However, the major problem of this work is that hypothesized benefits of PPRS are not backed up empirically, e.g. for section 5, and also the experimental section 6 seems unreliable at the current state. Due to my limited overview over current optimization literature, it is hard for me to judge the originality of this work and especially the proposed PPRS algorithm. Clarity: The paper is well written and structured. Theoretical concepts and theorems are described in an understandable manner. The figures and tables are also of high quality. Besides the challenges that arise from parallel computing, it seems, that one could easily implement the PPRS algorithm on the basis of the provided descriptions.

[Author Response · NeurIPS 2019]

We would like to thank all three reviewers for the close look given at the paper and the issues that were raised. You will
find below a detailed answer to each comment.

**About acceleration (REVIEWER #1)**    In the experimental section, we did not use acceleration for PPRS by referring
to Theorem 4 of Sec. 4.2 (the non-convex case) that does not use acceleration (*i.e.*, $\mu = 0$). For the sake of completeness,
we also tried the accelerated version of the algorithm (with $\mu = 0.99$), which did not improve the results. Hence, to
improve the readability of the figures, we decided not to show them and keep the (non-convex) theoretical version of
the algorithm. We will add a comment about this in the final version.

**Time computation in Fig. 3 (REVIEWER #1)**    The term "Time (normalized)" in the x-axis was indeed misleading.
We thus decided to replace it with "Parallel computation time" and state in the caption: "The parallel computation time
of each algorithm is computed through the formula $T(K + \Delta - 1)$, where $T$ is the number of iterations of an algorithm,
$K$ its number of samples per iteration and $\Delta$ the depth of the computation graph.". The time axis was indeed correctly
scaled to take into account that PPRS with $K = 200$ and $\Delta = 20$ has a computation time per iteration approximately
10 times bigger than any other method.

**Objectives of the experimental section (REVIEWER #2)**    The main focus of the paper is to provide a theoretical
analysis of this optimization problem, and experiments are only provided to underline the analysis. More specifically,
our aim is to show that some optimization problems are sufficiently difficult and non-smooth to require specialized
optimization methods (*e.g.*, random smoothing and PPRS). The practical impact and efficiency of PPRS and the
selection of its hyperparameters in practice is a very interesting line of research that we leave for future work.

**Distribution mechanism and relation to related works (REVIEWER #2)**    While pipeline-parallel optimization is
an abstraction that may be used for many different distribution setups, one of the main application is DL architectures
distributed on multiple GPUs (with memory limitations and communication bandwidths) by partitioning the model.
GPipe (with GD/AGD) may be seen as a special case of PPRS, with $\gamma, K = 0$ (*i.e.*, no randomized smoothing).

**PPRS on smooth functions and relevance to ML/DL(REVIEWER #2 AND #3)**    Note that, technically, neural
networks have non-smooth losses as soon as ReLU activation functions are used. However, the losses tend to be
relatively smooth in practice. As is, we do not believe that PPRS would improve on GD/AGD for smooth problems
(initial experiments seemed to support this intuition), and is more relevant for particularly hard optimization problems.
However, this does not mean that it could not be useful for ML, as: 1) many practical aspects of PPRS can yet be
improved. This should be the focus of follow-up work, as our work is of a more theoretical nature (see discussion in the
paragraph on the objectives of the experiments). 2) One may argue that the lack of efficient optimization methods for
hard optimization problems has drawn the ML community to more simple/smooth problems (e.g. using a quadratic
loss instead of the hinge loss, or particular DL architectures that are easy to learn), and easily available optimizers for
more difficult problems may enable researchers to discover novel architectures that might have not been learnable using
standard GD/AGD (see also the last sentence of Sec. 6).

**Robustness of the experimental results (REVIEWER #3)**    To assess the robustness of the approach with respect to
different optimization problems, we ran the attack on 100 images from CIFAR10 with the same set of hyperparameters.
Figure 1 shows the mean (and its standard deviation) of the minimized loss during training. The conclusions remain
unchanged.

Figure 1: Figure 3 from the paper to be updated.

[Meta-Review · NeurIPS 2019]

The reviewers agreed that this paper is a nice contribution to the literature and provides interesting and potentially useful convergence results in the framework of pipeline parallel optimization. The reviewers were impressed by the rebuttal and encourage the authors to incorporate the clarifications therein into the paper.